# Evaluation of therapeutic effect of oral Ursodeoxycholic Acid on indirect hyperbilirubinemia in term neonates undergoing phototherapy: A randomized controlled clinical trial

**Marjaneh Zarkesh[1], Sadroddin Mahdipour[1], Safoora Aghili[1], Atefeh Jafari[2], Seyyedeh Azadeh Hoseini Nouri[1], Afagh Hassanzadeh Rad[1], Maryam Ghalandari[3], Manijeh Tabrizi**[1]*

1 Pediatric Diseases Research Center, Guilan University of Medical Sciences, Rasht, Iran, 2 Department of Clinical Pharmacy, School of Pharmacy, Guilan University of Medical Sciences, Rasht, Iran, 3 PhD Student in Epidemiology, Shahid Sadoughi University of Medical Sciences, Yazd, Iran

* drs.tabrizi@gmail.com

## Abstract

### Introduction and aims

Phototherapy is the most common treatment modality of neonatal hyperbilirubinemia. We aimed to evaluate the therapeutic effect of oral Ursodeoxycholic Acid (UDCA) on indirect hyperbilirubinemia in term neonates undergoing phototherapy.

### Materials and methods

This randomized controlled clinical trial was performed on 106 full-term neonates with jaundice who were admitted to the neonatal ward of 17 Shahrivar Hospital in Rasht, Iran. The neonates were randomly assigned to two groups of intervention (10 mg/kg UDCA+phototherapy) and control (phototherapy alone). Total serum bilirubin (TSB) was measured at the time of admission, during first 12, 24, and 48 hours after admission and at the time of discharge. The duration of hospitalization and side effects were also assessed in both groups. IBM SPSS Statistics for Windows, version 20 was used to analyze the data.

### Results

Results showed that in the intervention group, 28 (52.8%) of neonates were boys with the mean age of 5.1±1.25 days. While, in the control group 29 (54.7%) of them were boys with the mean age of 5.19±2.26 days. Bilirubin levels in both groups decreased significantly after hospitalization (at 12, 24 and 48 hours) (P <0.001). The mean of bilirubin at 12, 24 and 48 hours in the intervention and control groups were 17.1, 13.2, 10.2 mg / dl and 17.1, 14.2 and 11.3 mg / dl, respectively. At the time of discharge, TSB in the former compared to the latter group was significantly reduced (7.74± 1.39 vs. 8.67±1.35) (P = 0.001). In addition, the

**Data Availability Statement:** "All data are available within the manuscript and/or Supporting Information files."

**Funding:** The authors received no specific funding for this work.

**Competing interests:** The authors have declared that no competing interests exist.

duration of hospitalization was considerably shorter in the intervention compared to the control group (P = 0.038) and no side effects were observed.

## Conclusions

Administering UDCA plus phototherapy reduced TSB and length of hospital stay with proper safety and efficacy. Therefore, it seems that this combination can be an appropriate treatment modality in neonatal hyperbilirubinemia.

## Introduction

Hyperbilirubinemia is one of the most common neonatal disorders, especially in the first week after birth, which occurs in half of term and most preterm infants [1]. Treatment modalities such as phototherapy, liver enzyme activity inducers, albumin, intravenous immunoglobulin (IVIG), and exchange transfusion are commonly used for neonatal jaundice. So far, few pharmacological agents such as metalloporphyrins, dipenicillamine, phenobarbital, clofibrate, and bile salts have been used clinically to treat neonatal hyperbilirubinemia [2].

Ursodeoxycholic Acid (UDCA) is a type of hydrophilic bile acid that accounts for about 4% of bile acids in humans and is synthesized by bacterial enzymes in the gallbladder from primary bile acid chenodeoxycholic acid. During oral administration, almost half of the drug, is absorbed in the intestine and enters hepatocytes through the portal circulation. UDCA significantly reduces the secretion of cholesterol into the bile [3]. As it is more hydrophilic than other bile acids, it gradually replaces hydrophobic acids that have accumulated in the bile during cholestasis, thus facilitating bile outflow from the liver/gallbladder. It also reverses the induction of programmed cell death (apoptosis) of indirect bilirubin on hepatocytes and nerve cells [4] that is associated with the protection of brain and liver tissue against indirect bilirubin.

It is noteworthy that pediatricians have to be assured about the safety profile of drugs used for neonates. Although some previous studies mentioned the positive effect of administering UDCA on neonatal hyperbilirubinemia [5–10], yet there is no consensus on its routine use. Therefore, it is not prescribed as an adjunct treatment in neonatal wards and further studies are needed. Based on the controversial results regarding the use of UDCA on hyperbilirubinemia and the shortage of evidence about its side effects, the aim of this study was to evaluate the therapeutic effect of oral UDCA plus phototherapy in the management of neonatal hyperbilirubinemia and to identify any side effects of this treatment. We hypothesized that administering UDCA plus phototherapy may have a significant effect on the change of TSB level from baseline as the primary outcome and the difference in length of hospitalization as the secondary outcome with no considerable side effect.

## Materials and methods

### Patients and settings

The present study is a paralleled randomized single-blinded controlled clinical trial that was performed from 10 April 2021 to 10 October 2021 for 6 months. In this study, 106 term neonates were admitted for at least 48 hours due to indirect hyperbilirubinemia in the neonatal ward of 17 Shahrivar Hospital in Rasht, Iran.

This study was approved by the ethics committee of the Vice-Chancellor of Research at Guilan University of Medical Sciences (Number: IR.GUMS.REC.1399.645, Date: 2021-03-03).

The clinical trial was registered in the Iranian registry of clinical trials (Number: IRCT20210201050199N1, Date: 2021-04-03). The authors confirm that all ongoing and related trials for this drug/intervention are registered. Written Informed consent letter was obtained from parents or guardians.

The inclusion criteria were 3 to 7 days of age, 2500 to 4000 grams birth weight, exclusive breastfeeding, 38 to 42 weeks of gestation, 14–20 mg/dL total serum bilirubin (TSB), and less than 1.5 mg/dL direct bilirubin. Neonates were excluded if there was ABO and Rh incompatibility, glucose 6 phosphate dehydrogenase deficiency (G6PDd), history of any neurological or chronic diseases, sepsis, diseases leading to hyperbilirubinemia (e.g., Crigler-Najjar and Gilbert's syndromes, hypothyroidism, and liver disease) and a history of maternal diabetes as well as parental dissatisfaction.

Eligible patients were randomly allocated by randomized blocking (or grouping). The following website "https://www.sealedenvelope.com" was used by a third party to produce the randomized list. Considering 27 blocks with the size of 4 and 6 possible variations (including AABB, ABAB, ABBA, BAAB, BABA, BBAA) were obtained. Then each patient received a specific code. Each code was placed in a separate sealed envelope. Finally an anonymous colleague assigned the eligible patients in the proper intervention and control groups after obtaining signed written informed consent. Parents/guardians were blinded to the groups enrolled.

The intervention group received 10 mg/kg/day oral UDCA divided into two doses plus phototherapy, which was dissolved in breast milk (Amin Pharmaceutical Company, Iran). The control group received phototherapy alone. The method of phototherapy was the same in both groups. Based on the AAP guideline [11], the acceptable bilirubin at discharge was considered as half of the exchange level for included neonates.

Sample size was indicated based on Shahramian et al. [5]. With the confidence interval of 95% and 80% test power, the sample size was equal to 48 neonates in each group. Considering the drop rate of 10% during the study, 53 neonates in each group were required.

$$n_2 = \frac{(p_1(1 - p_1) + p_2(1 - p_2))\left(Z_{1 - \frac{\alpha}{2}} + Z_{1 - \beta}\right)^2}{(p_1 - p_2)^2}$$

$$\frac{(0.72(1 - 0.72_1) + 0.45(1 - 0.45))(1.96 + 0.84)^2}{(0.72 - 0.45)^2}$$

$n_2 = 48$
10% drop rate = 5
N = 53 in each group

## Data gathering

Data were gathered by a checklist including baseline and post randomization variables. Baseline characteristics were age, sex, gestational age, and baseline blinrubin level. Also, bilirubin level at 12, 24, and 48 hours after admission and side effects including nausea, vomiting, constipation, bloating, and skin rash were assessed as post randomization variables in the two groups. To measure serum bilirubin, arterial blood samples were taken from the neonates and transferred to the single laboratory and was performed by colorimetric assay using 2 and 4-dichloroaniline (DCA) in a 3500BT device. We considered the change of TSB level from baseline as the primary outcome and the difference in length of hospitalization as the secondary outcome.

## Statistical analysis

Data were analyzed using IBM SPSS Statistics for Windows, version 20 (IBM Corp., Armonk, N.Y., USA). Data were reported by frequency, percent, mean, and standard deviation. Qualitative variables were also described based on number and percent. At first, the normality of the data was evaluated using the Kolmogorov-Smirnov test. Then, independent T and chi-square tests were used to compare the groups in terms of different variables. It is noteworthy that in case of non-normal data, Mann-Whitney U test and if data did not fulfill the necessary condition for chi-square, Fisher's exact test was used. We used the Friedman test to compare the quantitative non-normal data. P-value less than 0.05 was considered significant. The investigation was performed based on the intention to treat analysis.

## Results

In the present study, 198 admitted neonates were assessed for eligibility from 10 April 2021 to 10 October 2021 for 6 months. Among them, 106 were enrolled in two groups of 53 neonates (Fig 1). Comparing groups showed no significant difference regarding sex (P = 0.846), age (P = 0.464), and weight (P = 0.459) (Table 1).

According to the results, a statistically significant difference was observed between the total bilirubin levels at the time of admission and discharge in both groups (P <0.001).

As Table 2 shows, TSB as the primary outcome at the time of admission in the intervention group was not statistically significant compared to the control group (P <0.198). However, TSB at discharge in the former group significantly decreased compared to the latter (P = 0.001).

In addition, bilirubin levels in both groups were measured during 12, 24, and 48 hours after admission, and the results were obtained by Friedman test. We intended to investigate the difference in bilirubin levels in these three time periods, the neonates studied at this stage were 49 in the intervention and 43 in the control group. Therefore, 14 participants were excluded due to loss to follow-up. The intragroup analysis indicated that TSB in both groups had significant decrease during 12, 24, and 48 hours after hospitalization (P <0.001). Although there was no significant difference between the two groups at 12 and 48 hours after hospitalization (P>0.05), significant difference was noticed at 48 hours after hospitalization (P = 0.009) (Table 3). Fig 2 depicts the trend of bilirubin levels of all, intervention, and control groups from baseline to discharge.

Using the Mann-Whitney U test, the length of hospital stay as the secondary outcome was significantly reduced in the intervention compared to the control group (P = 0.038) (Table 4).

No side effects including nausea, vomiting, constipation, bloating, and skin rash were observed in the two groups.

## Discussion

Phototherapy is a cheap, simple, non-invasive method with appropriate effectiveness. This treatment has side effects such as changes in circadian rhythm, dehydration, hypocalcemia, skin rash, and retinal damage [12]. Thus; it seems that applying other methods of treatment in combination with phototherapy that can reduce the duration and increase its efficacy, can be useful. The findings of this study showed that the use of oral UDCA significantly reduced the amount of total bilirubin at the time of discharge which was in line with the study by Honar et al. They conducted a double-blind randomized clinical trial on 80 neonates with jaundice. They were randomly divided into two groups of intervention (receiving 10 mg/kg/ day UDCA in addition to phototherapy), and control (receiving only phototherapy). Results suggested that administering UDCA had a positive therapeutic effect on indirect hyperbilirubinemia,

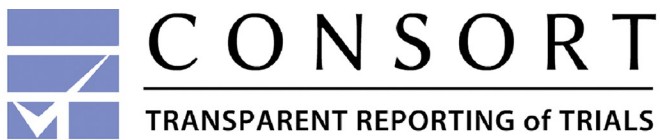

```
                          Assessed for eligibility
                                 (n=198)
```

```
                                              Excluded (n=92)
                                                  Prematurity (n=50)
                                                  History of maternal diabetes (n=20)
                                                  Small for gestational age (n= 15)
                                                  Seizure (n= 4 )
```

```
                              Randomized
                                (n=106)
```

```
        UDCA + phototherapy                    Phototherapy
             (n=53)                               (n=53)
```

```
  Excluded (n=4)                                         Excluded (n=10)
  Lost to follow-up due to                                Lost to follow-up due to
  unwillingness                                           unwillingness
```

```
              Analyzed                            Analyzed
               (n=49)                              (n=43)
```

**Fig 1. Flow chart of participants.**

reduced the time for phototherapy, and shortened the length of hospital stay [6]. In addition, results of Jafari et al. who assessed 96 term neonates with hyperbilirubinemia indicated that oral administration of UDCA at a dose of 10 mg / kg with phototherapy significantly reduced TSB compared with the phototherapy alone. They reported the same efficacy for the dose of 10

**Table 1. Baseline demographic characteristics in the intervention and control groups.**

| Variables | Intervention group (n = 53) | Control group (n = 53) | P Value |
|---|---|---|---|
| age(yr) mean ± SD | 5.10 ± 1.25 | 5.18 ± 2.26 | 0.464* |
| weight(gr) mean ± SD | 3263.02 ± 364.81 | 3207.74 ± 400.17 | 0.459** |
| Sex (male) n(%) | 28 (52.8%) | 29 (54.7%) | 0.846*** |

* Mann-Whitney U test.

** Independent samples T-test.

*** Chi-Squared test.

versus 20 mg / kg of UDCA [7]. Akefi et al. performed a randomized clinical trial on 220 neonates receiving 10 mg / kg oral UDCA twice daily plus phototherapy and the control group. They found that the mean 12 hours TSB decreased by 2.70 mg / dl in the control versus 3.70 mg / dl in the intervention group. In addition, at 24 hours, the mean TSB diminished by 5.22 mg / dl in the control versus 6.54 mg / dl in the intervention group (p = 0.01) [8].

In the present study, we compared TSB at the beginning of hospitalization and at the time of discharge between two groups. TSB at discharge was significantly reduced in the UDCA group. Although TSB at admission in both groups was about 17.1 mg / dl, but at discharge, it was significantly lower in the intervention rather than the control group. In the previous studies, bilirubin levels have been measured at regular intervals with the same interventions. Although Jafari et al. who compared groups at 8, 12, 24, 36, and 48 hours of hospitalization showed decreased bilirubin level in both groups, the difference was statistically significant only at 8 [7]. Consistent with the results of the present study, in the study by Honar et al., bilirubin level showed significant decrease in all three time points of 12, 24, and 48 hours in both groups [6]. In line with these results, Gharehbaghi et al. compared three groups of neonates with hyperbilirubinemia receiving oral UDCA at a dose of 7.5 mg / kg with phototherapy (group A), oral UDCA at a dose of 5 mg/ kg with phototherapy (group B) and phototherapy alone (group C). Results proposed that at the time of discharge, bilirubin decrement was significantly higher in group A (4.74 mg / dl) than in groups B (4.75 mg/dl), and C (7.88 mg/dL) (P< 0.001) [9].

Based on the findings of our study, the mean duration of hospitalization in the UDCA group was significantly lower than the control group. Consistent with our results, previous studies mentioned a significant difference between mean duration of phototherapy between two groups as well [6,7,10].

**Table 2. Comparing bilirubin level at admission and discharge in the intervention and control groups.**

| Variables | Intervention group (n = 53) | Control group (n = 53) | P Value** |
|---|---|---|---|
| Bilirubin at the admission (mg/dl) SD ± mean | 17.03±1.81 | 17.49±1.92 | 0.198 |
| Bilirubin at the discharge (mg/dl) SD ± mean | 7.74±1.39 | 8.67±1.35 | <0.001 |
| P-value* | <0.001 | <0.001 | |

*Wilcoxon signed-rank test.

** Mann-Whitney U test.

**Table 3. Inter and intra group analyses of bilirubin level in 12, 24, and 48 hours after hospitalization.**

| Groups | Bilirubin at 12 hours after hospitalization (mg/dl) mean ± SD | Bilirubin at 24 hours after hospitalization (mg/dl) mean ± SD | Bilirubin at 48 hours after hospitalization (mg/dl) mean ± SD | P-value* |
|---|---|---|---|---|
| Intervention group (n = 49) | 17.68±1.85 | 13.73±2.6 | 10.09±2.48 | <0.001 |
| Control group (n = 43) | 17.30±1.83 | 14.24±2.16 | 11.83±1.33 | <0.001 |
| P- value** | 0.351 | 0.271 | 0.009 | |

*Friedman test.

*** Mann-Whitney U test.

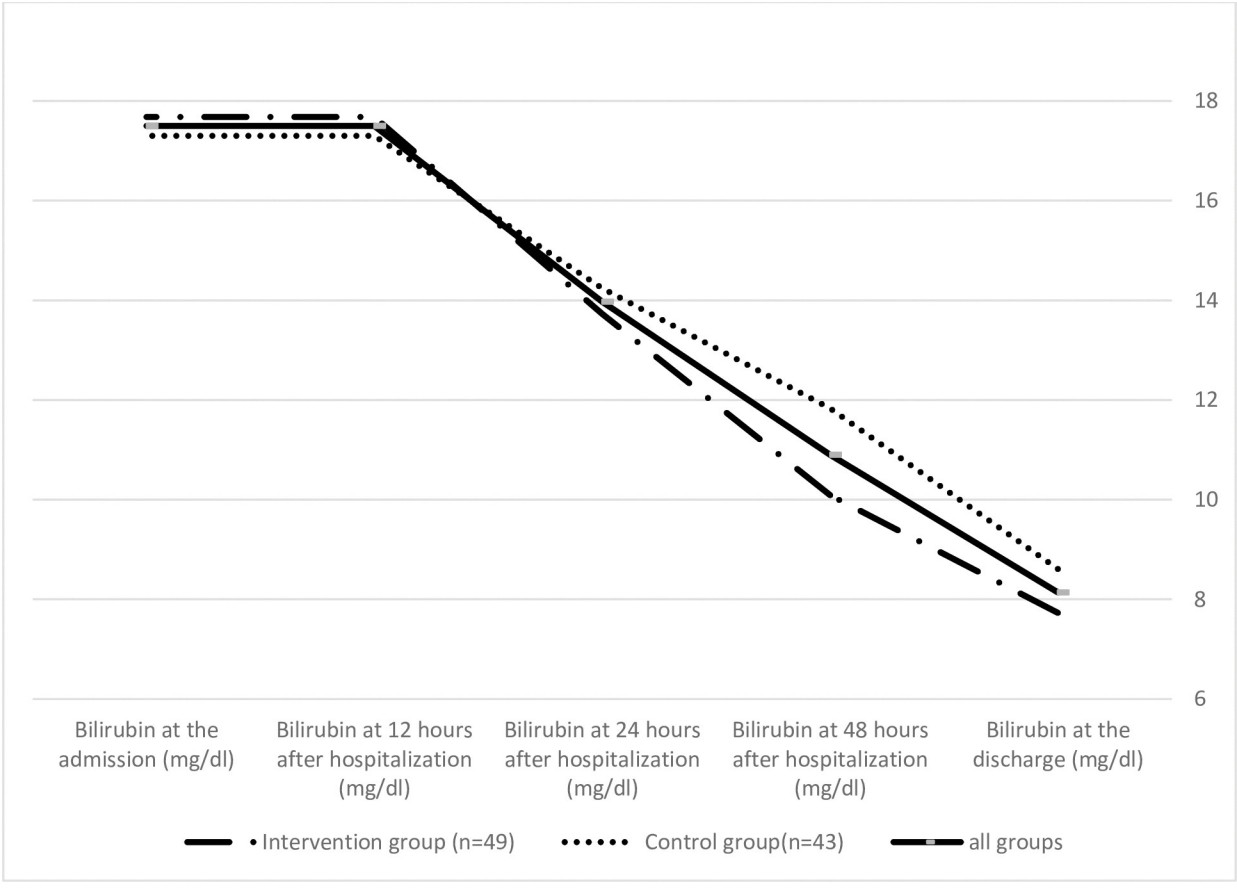

**Fig 2. The trend of bilirubin levels of all, intervention, and control groups from baseline to discharge.**

**Table 4. Comparing duration of hospitalization in the intervention and control groups.**

| Variable | Intervention group (n = 53) | Control group (n = 53) | P-Value* |
|---|---|---|---|
| Duration of hospitalization (hours) SD ± mean | 38.49±16.53 | 43.52±14.56 | 0.038 |

* Mann-Whitney U test.

Gharehbaghi et al. also revealed that the length of hospital stay was reduced and more than 70% of infants receiving the dose of 7.5 mg / kg of UDCA were hospitalized for less than one day, but in the group receiving the dose of 5 mg / kg and the control group, 42.5% and 17.5% of infants were hospitalized less than one day [9]. In contrast to the above studies, Akefi et al., noted no significant difference between the two groups receiving oral UDCA at a dose of 10 mg / kg with phototherapy and phototherapy alone regarding the mean duration of phototherapy (34.6 versus 35.5 hours, respectively (p = 0.63)) [8]. These results indicated the need for a more detailed study on the effectiveness of oral UDCA in reducing the time required for phototherapy in the clinic.

Also lack of any possible side effects including nausea, vomiting, constipation, and skin rash in the intervention group suggested an acceptable and safe profile of the drug. Consistent with these results, several human and animal studies have confirmed the safety and efficacy of UDCA in both adults and children [8,13–17].

We have one main limitation that should be considered in further studies. It would be better if we could compare term neonates with preterm ones. Besides, comparing term neonates with and without risk factors such as ABO & Rh incompatibilities, G6PDd, and infants of diabetic mothers might be considerably informative.

## Conclusions

Based on the findings, administering UDCA plus phototherapy reduced TSB and length of hospital stay with appropriate safety and efficacy. Therefore, it seems that this combination can be an appropriate treatment modality in neonatal hyperbilirubinemia. Besides, further studies can be recommended to evaluate the efficacy of UDCA on outpatient management of prolonged neonatal jaundice.

## Supporting information

**S1 Checklist. Consort checklist: CONSORT 2010 Checklist dr aghili.**
(DOC)

**S1 Data. Data set: Dr aghili–analysis.**
(XLSX)

**S1 File. Persian protocol: Trials persian dr aghili.**
(PDF)

**S2 File. English protocol: Trials english dr aghili.**
(PDF)

**S3 File. Persian and English protocol summary: Protocol dr aghili revised.**
(DOCX)

**S4 File. Consort flow diagram: CONSORT 2010 flow diagram dr aghili.**
(DOC)

## Acknowledgments

This was the proposal accepted by the Vice-Chancellor of Research at Guilan University of Medical Sciences. It was the thesis of the Dr. Safoora Aghili resident of pediatrics (third Author). We appreciate all colleagues and parents/guardians for their valuable cooperation.

## Author Contributions

**Conceptualization:** Marjaneh Zarkesh, Sadroddin Mahdipour, Safoora Aghili, Atefeh Jafari, Seyyedeh Azadeh Hoseini Nouri, Afagh Hassanzadeh Rad, Manijeh Tabrizi.

**Data curation:** Marjaneh Zarkesh, Sadroddin Mahdipour, Afagh Hassanzadeh Rad, Manijeh Tabrizi.

**Formal analysis:** Marjaneh Zarkesh, Sadroddin Mahdipour, Afagh Hassanzadeh Rad.

**Funding acquisition:** Sadroddin Mahdipour, Afagh Hassanzadeh Rad.

**Investigation:** Marjaneh Zarkesh, Safoora Aghili, Atefeh Jafari, Maryam Ghalandari.

**Methodology:** Atefeh Jafari, Maryam Ghalandari, Manijeh Tabrizi.

**Project administration:** Sadroddin Mahdipour.

**Resources:** Sadroddin Mahdipour, Afagh Hassanzadeh Rad.

**Software:** Sadroddin Mahdipour, Safoora Aghili, Maryam Ghalandari.

**Supervision:** Marjaneh Zarkesh, Sadroddin Mahdipour, Safoora Aghili, Atefeh Jafari, Seyyedeh Azadeh Hoseini Nouri, Manijeh Tabrizi.

**Validation:** Marjaneh Zarkesh, Sadroddin Mahdipour, Safoora Aghili, Seyyedeh Azadeh Hoseini Nouri, Afagh Hassanzadeh Rad, Manijeh Tabrizi.

**Visualization:** Marjaneh Zarkesh, Sadroddin Mahdipour, Safoora Aghili, Seyyedeh Azadeh Hoseini Nouri, Afagh Hassanzadeh Rad, Manijeh Tabrizi.

**Writing – original draft:** Marjaneh Zarkesh, Sadroddin Mahdipour, Safoora Aghili, Atefeh Jafari, Seyyedeh Azadeh Hoseini Nouri, Afagh Hassanzadeh Rad, Maryam Ghalandari, Manijeh Tabrizi.

**Writing – review & editing:** Marjaneh Zarkesh, Sadroddin Mahdipour, Safoora Aghili, Atefeh Jafari, Seyyedeh Azadeh Hoseini Nouri, Afagh Hassanzadeh Rad, Maryam Ghalandari, Manijeh Tabrizi.

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
