## [Decision Letter · Decision Letter 0]

4 May 2022

PONE-D-22-07805Evaluation of therapeutic effect of oral Ursodeoxycholic Acid on indirect hyperbilirubinemia in term neonates undergoing phototherapy: A randomized controlled clinical trial.PLOS ONE

Dear Dr. Tabrizi,

Thank you for submitting your manuscript to PLOS ONE. After careful consideration, we feel that it has merit but does not fully meet PLOS ONE’s publication criteria as it currently stands. Therefore, we invite you to submit a revised version of the manuscript that addresses the points raised during the review process.

We look forward to receiving your revised manuscript.

Kind regards,

Wan-Long Chuang, M.D., Ph.D.

Academic Editor

PLOS ONE

Journal Requirements:

Reviewers' comments:

Reviewer's Responses to Questions

**Comments to the Author**

1. Is the manuscript technically sound, and do the data support the conclusions?

Reviewer #1: Partly

Reviewer #2: No

Reviewer #3: No

2. Has the statistical analysis been performed appropriately and rigorously? 

Reviewer #1: Yes

Reviewer #2: No

Reviewer #3: No

3. Have the authors made all data underlying the findings in their manuscript fully available?

Reviewer #1: Yes

Reviewer #2: No

Reviewer #3: Yes

4. Is the manuscript presented in an intelligible fashion and written in standard English?

Reviewer #1: Yes

Reviewer #2: Yes

Reviewer #3: Yes

5. Review Comments to the Author

Reviewer #1: The authors conducted this study and aimed to investigate the effect of ursodeoxycholic acid on indirect hyperbilirubinemia in term neonates undergoing phototherapy. They concluded that UDCA plus phototherapy reduced total serum bilirubin and length of hospital stay with proper safety and efficacy. In general, the study design is adequate and well conducted with believable results. However, UDCA in the treatment of hyperbilirubinemia in neonates had been studied before as authors mentioned in the discussion. There are no novel findings in the present study. Besides, authors should discuss more regarding the discrepancies between the present and previous studies. Other comments list below.

1. In conclusions of main text and Abstract. “Administering UCDA plus phototherapy reduced TSB….” It should be “UDCA”, not “UCDA”.

2. Is the effect of UDCA different in different subgroup of neonates like male/female. Authors should perform subgroup analysis to understand whether similar effect of UDCA in different subgroups.

3. A table of baseline demographics should be added in the main text.

4. Is the total serum bilirubin level significantly different between intervention and control groups at 12, 24, and 48 hours? Authors may also add a figure drawing the decline of bilirubin levels of all/intervention/control groups from baseline to discharge.

5. The results showed combined UDCA therapy had significantly lower TSB level at discharge and shorter duration of hospitalization. Authors should further adjust UDCA therapy with other associated factors to confirm the independent association of UDCA therapy.

Reviewer #2: This a study evaluating the therapeutic effect of oral Ursodeoxycholic Acid plus phototherapy in the management of neonatal hyperbilirubinemia.

1. It would be good if the authors expanded on the content in the introduction section. To include rationale for the proposed RCT. Also it would be good to include more studies that have also looked at similar effects and why this study is so important.

2. The randomisation section is very brief, can the authors elaborate on who was blinded and also more details with regards to how allocation concealment was ensured. Its not clear from the sentence “According to the list obtained from the software and given a block size of 4, each case was placed in a separate envelop and its lid was closed and given to a third party”

3. Additionally can the authors explicitly say who was blinded. From reading line 87, it implies Eligible patients were blinded, presumably do you mean parents/guardians since these were neonates who would not even assimilate what is going on.

4. Can the authors give more details on the sample size, as it does not contain the effect size of the outcome and so impossible to replicate.

5. The methods needs to include details and definitions of outcomes. i.e saying “we considered TSB level as the primary outcome” is not enough as you need to define what exactly you looking for. For example is it change of TSB from baseline. Same comment for secondary outcome, i.e. length of hospitalization.

6. A minor comment, when the authors mention data collected. For example makes it clear to say what was collected at baseline and what was collected post randomisation.

7. The last paragraph in the introduction should make clear the primary (i.e effect of xxxx on total serum bilirubin) and secondary objectives (length of hospitalization) and safety objectives.

8. The statistical analysis is very vague on the analysis of the specific outcomes.

9. As this is an RCT, there should be no formal comparisons made between baseline characteristics as any differences observed will be by chance.

10. An observation is that the attached protocol is a summary and should be the full protocol attached. As the study is complete now, therefore would serve useful information to accompany the manuscript.

Reviewer #3: The authors aimed to evaluate the therapeutic effect of oral ursodeoxycholic acid (UDCA) on indirect hyperbilirubinemia in term neonates undergoing phototherapy in a randomized controlled clinical trial. They found that at the time of discharge, total serum bilirubin (TSB) level in the combined treatment compared to the latter group was significantly reduced (7.74± 1.39 vs. 8.67±1.35). In addition, the duration of hospitalization was considerably shorter in the intervention compared to the control group. They concluded that combined UCDA plus phototherapy can be an appropriate treatment modality in neonatal hyperbilirubinemia. The comments were as followed.

1. There were no any creative viewpoints in this study compared to previous published data with similar study design and results.

2. All participants who are randomized in an intention-to-treat analysis should be included in the statistical analysis.

3. Table 2. The analysis of bilirubin level in 12, 24, and 48 hours after hospitalization should be compared between intervention and control group.

4. How did the authors explain no differences in TSB level at 12, 24 and 48 hours, but significantly reduction at the time of discharge between intervention and control groups?

6. PLOS authors have the option to publish the peer review history of their article (what does this mean?). If published, this will include your full peer review and any attached files.

Reviewer #1: No

Reviewer #2: No

Reviewer #3: No

---

## [Author Response · Author response to Decision Letter 0]

7 Jul 2022

Dear Editor in chief,

We confirm the following proposed statement accurate and suitable to appear alongside our manuscript.

“All data are available within the manuscript and Supporting Information files.”

With regards,

Manijeh Tabrizi

---

## [Decision Letter · Decision Letter 1]

10 Aug 2022

Evaluation of therapeutic effect of oral Ursodeoxycholic Acid on indirect hyperbilirubinemia in term neonates undergoing phototherapy: A randomized controlled clinical trial.

PONE-D-22-07805R1

Dear Dr. Tabrizi,

We’re pleased to inform you that your manuscript has been judged scientifically suitable for publication and will be formally accepted for publication once it meets all outstanding technical requirements.

Kind regards,

Wan-Long Chuang, M.D., Ph.D.

Academic Editor

PLOS ONE

Additional Editor Comments (optional):

Reviewers' comments:

Reviewer's Responses to Questions

**Comments to the Author**

1. If the authors have adequately addressed your comments raised in a previous round of review and you feel that this manuscript is now acceptable for publication, you may indicate that here to bypass the “Comments to the Author” section, enter your conflict of interest statement in the “Confidential to Editor” section, and submit your "Accept" recommendation.

Reviewer #1: All comments have been addressed

Reviewer #2: All comments have been addressed

Reviewer #3: All comments have been addressed

2. Is the manuscript technically sound, and do the data support the conclusions?

Reviewer #1: (No Response)

Reviewer #2: Yes

Reviewer #3: Partly

3. Has the statistical analysis been performed appropriately and rigorously? 

Reviewer #1: (No Response)

Reviewer #2: Yes

Reviewer #3: Yes

4. Have the authors made all data underlying the findings in their manuscript fully available?

Reviewer #1: (No Response)

Reviewer #2: Yes

Reviewer #3: Yes

5. Is the manuscript presented in an intelligible fashion and written in standard English?

Reviewer #1: (No Response)

Reviewer #2: Yes

Reviewer #3: Yes

6. Review Comments to the Author

Reviewer #1: (No Response)

Reviewer #2: (No Response)

Reviewer #3: The authors aimed to evaluate the therapeutic effect of oral ursodeoxycholic acid (UDCA) on indirect hyperbilirubinemia in term neonates undergoing phototherapy in a randomized controlled clinical trial. They concluded that combined UCDA plus phototherapy can be an appropriate treatment modality in neonatal hyperbilirubinemia. In general, this is an interesting topic and a clearly written paper. Also, the authors have well responded to the reviewer’s suggestions.

7. PLOS authors have the option to publish the peer review history of their article (what does this mean?). If published, this will include your full peer review and any attached files.

Reviewer #1: No

Reviewer #2: No

Reviewer #3: No

---

## [Editor Report · Acceptance letter]

19 Sep 2022

PONE-D-22-07805R1 

Evaluation of therapeutic effect of oral Ursodeoxycholic Acid on indirect hyperbilirubinemia in term neonates undergoing phototherapy: A randomized controlled clinical trial. 

Dear Dr. Tabrizi:

I'm pleased to inform you that your manuscript has been deemed suitable for publication in PLOS ONE. Congratulations! Your manuscript is now with our production department. 

Kind regards, 

on behalf of

Dr. Wan-Long Chuang 

Academic Editor

PLOS ONE